# Spin Glass State and Griffiths Phase in van der Waals Ferromagnetic Material Fe_5_GeTe_2_

**DOI:** 10.3390/nano15010019

**Published:** 2024-12-27

**Authors:** Jiaqi He, Yuan Cao, Yu Zou, Mengyuan Liu, Jia Wang, Wenliang Zhu, Minghu Pan

**Affiliations:** School of Physics and Information Technology, Shaanxi Normal University, Xi’an 710119, China; jiaqihe@snnu.edu.cn (J.H.); cao_yuan@snnu.edu.cn (Y.C.); zouyu@snnu.edu.cn (Y.Z.); liumy@snnu.edu.cn (M.L.); tnsiwtpsir@163.com (J.W.); wlzhu@snnu.edu.cn (W.Z.)

**Keywords:** Fe_5_GeTe_2_, spin glass state, Griffith phase

## Abstract

The discovery of two-dimensional (2D) van der Waals ferromagnetic materials opens up new avenues for making devices with high information storage density, ultra-fast response, high integration, and low power consumption. Fe_5_GeTe_2_ has attracted much attention because of its ferromagnetic transition temperature near room temperature. However, the investigation of its phase transition is rare until now. Here, we have successfully synthesized a single crystal of the layered ferromagnet Fe_5_GeTe_2_ by chemical vapor phase transport, soon after characterized by X-ray diffraction (XRD), DC magnetization M(T), and isotherm magnetization M(H) measurements. A paramagnetic to ferromagnetic transition is observed at ≈302 K (*T*_C_) in the temperature dependence of the DC magnetic susceptibility of Fe_5_GeTe_2_. We found an unconventional potential spin glass state in the low-temperature regime that differs from the conventional spin glass states and Griffiths phase (GP) in the high-temperature regime. The physical mechanisms behind the potential spin glass state of Fe_5_GeTe_2_ at low temperatures and the Griffith phase at high temperatures need to be further investigated.

## 1. Introduction

In 2004, Andre Geim et al. reported that graphite can be exfoliated into several layers or even a single layer by mechanical exfoliation processes [1]; this atomic-thick graphite layer is called graphene. The discovery of graphene opened widespread interest in 2D materials and has offered a promising paradigm for the development of low-dimensional spintronic devices [2]. In particular, the spin–orbit coupling of 2D magnetic materials is the basis for potential applications of new low-dimensional spintronic devices, which has made the magnetoelectric coupling of spintronic devices possible [3,4,5]. Subsequent doping of these materials with magnetic elements resulted in the emergence of magnetic properties. Nevertheless, the magnetism observed in these materials is not intrinsic. The Mermin–Wagner theorem states that a one- or two-dimensional isotropic spin-s Heisenberg model with finite exchange interaction can be neither ferromagnetic nor antiferromagnetic at any nonzero temperature [6]. However, large magnetic anisotropy is believed to suppress random spin reorientation induced by thermal fluctuations and provides the existence of two-dimensional magnetism. In 2017, C. Gong et al. reported that 2D van der Waals magnet Cr_2_Ge_2_Te_6_ can maintain long-range magnetic order even at an atomic-scale thickness [7]. B. Huang et al. reported that 2D ferromagnetism in exfoliated monolayer CrI_3_ [8]. However, their potential applications remain constrained by the relatively low magnetic transition temperature. Soon, Fe_3_GeTe_2_, an exfoliable vdW magnet, was discovered, with a reasonably high *T*_C_ of about 220–230 K [9]. It exhibits robust 2D ferromagnetism with strong perpendicular anisotropy [10]. Furthermore, by applying an ionic gate, the *T*_C_ of thin flakes can be raised to room temperature, much higher than the bulk *T*_C_ [11].

Fe_5-x_GeTe_2_ is a tourmaline ferromagnet with a hexagonal crystal structure [12], which is noteworthy due to its very high Curie temperature (*T*_C_ = 270–310 K), and its ferromagnetic behavior persists near room temperature in flakes thinned down to four unit cells thick (12 nm) [13]. The magnetic properties of Fe_5_GeTe_2_ have received much attention recently. B. W. Casas et al. reported that merons and anti-merons in Fe_5−x_GeTe_2_ coexist with Néel skyrmionic states over a broad range of temperatures and demonstrated that the topological Hall effect (THE) even at room temperature [14]. C. Zhang et al. reported the unconventional helicity modulation behavior of magnetic sigmoid in Fe_5−x_GeTe_2_ nanosheets [15]. By employing an electric field modulation technique, M. Tang et al. demonstrated that the magnetic anisotropy properties of Fe_5_GeTe_2_ display a pronounced dependence on both temperature and sample thickness [16]. However, there is an absence of research in the field of the magnetic phase transition of Fe_5_GeTe_2_.

The phenomenon of spin-glass (SG) behavior was initially identified and postulated during the investigation of dilute magnetic alloys, later observed in certain chalcogenide oxides [17]. Once the sample has reached the spin-frozen state, the magnetic moments within the system are frozen in a disordered distribution in all directions. Following the random freezing of the spin in the SG, the magnetic moment is kept in a fixed direction and is unable to rotate freely. Furthermore, the ferromagnetic–paramagnetic transition refers to the change in the magnetic state of a material at a specific temperature, known as the Curie temperature (*T*_C_). At temperatures below the Curie temperature, the material displays ferromagnetic behavior, whereby the magnetic moments of the constituent atoms or molecules are parallelly aligned spontaneously. At temperatures above the Curie point, the material undergoes a transition to a paramagnetic state, wherein the magnetic moments are randomly oriented. Both spin-glass states and ferromagnetic–paramagnetic transitions are the fundamental issues for pursuing two-dimensional room-temperature ferromagnetic materials [11].

In this work, we study the magnetic phase transitions in Fe_5_GeTe_2_ single crystal by utilizing a physical property measurement system (PPMS), revealing two distinct, intriguing magnetic phase transitions: paramagnetic to ferromagnetic at *T*_C_~302 K and potential spin-glass transition at *T*_f_~98 K. In the low-temperature region, Fe_5_GeTe_2_ crystal is in a potential spin glass state. In the high-temperature region, Fe_5_GeTe_2_ exhibits non-linear magnetization due to the Griffiths phase. The physical mechanisms behind the potential spin glass state of Fe_5_GeTe_2_ at low temperatures and the Griffith phase at high temperatures need to be further investigated.

## 2. Sample Preparation and Characterization

### 2.1. Single-Crystal Growth

High-quality Fe_5_GeTe_2_ single crystals were successfully synthesized by the chemical vapor transport method, using iodine as the transporting agent. The mixture of Fe (Aladdin, 99.9%), Ge (Alfa Aesar, 99.999%), and Te (Alfa Aesar, 99.999%) with the ratio of 6:1:2 was sealed into an evacuated quartz ampoule. The ampoule was then heated to create a temperature gradient with 800–1000 °C at the hot end and 600–800 °C at the cold end of the quartz ampoule for 7–20 days to grow the single crystal.

Fe_5_GeTe_2_ crystallizes in a trigonal structure (space group R3m, No. 160) [18], comprising layered building blocks that are stacked along the c-axis, as illustrated in Figure 1a; each block contains a Fe-Ge slab comprising three distinct Fe atom sites (Fe1, Fe2, and Fe3, as illustrated in Figure 1a), with two layers of Te atoms positioned above and below the Fe-Ge slab, respectively. It is noteworthy that the additional Fe1 and Fe3 layers serve to distinguish the structural and magnetic properties of Fe_5_GeTe_2_ from those of Fe_3_GeTe_2_ [19].

### 2.2. X-ray Diffraction Measurements

The X-ray diffraction (XRD) pattern of as-grown Fe_5_GeTe_2_ single crystal at room temperature is shown in Figure 1b. During the XRD measurement, the X-rays were incident on a flat surface of the crystal, resulting in diffraction peaks exclusively corresponding to the (00l) planes. It indicates the single-crystalline nature of the material, and the inset of Figure 1b shows the photo of Fe_5_GeTe_2_, further revealing that the planes of the crystal represent the *ab* plane. Our XRD results agree well with the theoretical values, as shown in the table (Appendix A). Self-intercalation of the transition metal ion has been reported in vdW materials, for example, in CrTe_2_ [20]. Since the *c*-axis lattice constant of a Fe-intercalated Fe_5_GeTe_2_ single crystal will be greatly enlarged due to the existence of the intercalated layer, its X-ray diffraction pattern should be dramatically different from the standard Fe_5_GeTe_2_. Our XRD measurement rules out the existence of self-intercalation of the transition metal.

### 2.3. Scanning Electron Microscope Measurements

The scanning electron microscopy (SEM) analysis reveals that Fe_5_GeTe_2_ single crystal exhibits a layered structure with a relatively uniform morphology shown in Figure 1c. Most of the samples present in the form of hexagonal flake materials featured a smooth surface. This characteristic is highly favorable for the subsequent mechanical exfoliation or liquid-phase exfoliation processes to obtain few-layer samples.

### 2.4. Energy-Dispersive X-ray Spectroscopy Measurements

To investigate the elemental composition and chemical homogeneity of Fe_5_GeTe_2_, energy-dispersive X-ray (EDX) spectroscopy has been performed on the Fe_5_GeTe_2_ single crystal. As shown in Figure 1d,e, spots in the area on the surface of the crystal were measured. The values were finally averaged, which are presented in Appendix A, showing the compositions as Fe:Ge:Te = 5.14:1.00:2.39 without other impurities and matching the earlier report [19].

## 3. X-ray Photoemission Spectroscopy Measurements

The surface chemical composition and core-level binding energy of the Fe_5_GeTe_2_ single crystal were measured by utilizing X-ray photoemission spectroscopy (XPS). The XPS results for Fe, Ge, and Te in the Fe_5_GeTe_2_ single crystal are presented in Figure 2. As shown in Figure 2, Fe 2*p* spectra, Fe(III) doublets (Fe 2*p*_1/2_, Fe 2*p*_3/2_), and Fe(II) doublets (Fe 2*p*_1/2_, Fe 2*p*_3/2_) could be observed with their resonance transition (satellite peaks) at their corresponding binding energies. Fe^3+^ with binding energies of Fe 2*p*_3/2_ at 712.10 eV and Fe 2*p*_1/2_ at 725.56 eV is the dominant species in Fe_5_GeTe_2_, which becomes the clear evidence for the high-spin state of Fe in their ferromagnetic state. Furthermore, the presence of Fe^2+^ 2*p*_3/2_ (710.56 eV) and Fe^3+^ 2*p*_3/2_ (712.10 eV) peaks suggests that Fe exists in ionic form with a mixed valence of +2 and +3 in Fe_5_GeTe_2_. The analysis of the Ge 3*d*_5/2_ and Ge 3*d*_3/2_ peaks in Figure 2 reveals that Ge is present as both elemental Ge and Ge (IV) species in the Fe_5_GeTe_2_ single crystal; only the latter is used to determine the composition. Similarly, the Te 3*d* spectrum exhibits two peaks at a binding energy of 576.42 eV and 586.83 eV, corresponding to the 3*d*_3/2_ and 3*d*_5/2_ levels of the Te^4+^ oxidation state, while peaks at approximately 572.98 eV and 583.32 eV are attributed to metallic tellurium (Te^0^) in Figure 2. The oxidation of the surface atoms of Fe_5_GeTe_2_ upon exposure to air is suggested by the oxide peaks observed in all species [21]. Element Ge is nonmagnetic, which will not affect the measured magnetism, as reported in Fe_4_GeTe_2_ [21] and Fe_3_GeTe_2_ [22].

## 4. Magnetization Measurements

Figure 3 shows the temperature dependence of magnetization, the corresponding derivatives, and inverse magnetic susceptibility for the Fe_5_GeTe_2_ single crystal under an applied magnetic field of 3000 Oe parallel to the *c* axis. The temperature dependence of the DC magnetization was investigated for Fe_5_GeTe_2_ under zero-field-cooling (ZFC) and field-cooling (FC) conditions in a magnetic field of 3000 Oe applied along the crystallographic *c*-axis within a temperature range of 2–350 K, as shown in Figure 3a. The Fe_5_GeTe_2_ was initially cooled from 350 K to 2 K in the absence of an applied magnetic field. Subsequently, an applied magnetic field of 3000 Oe was introduced at 2 K, and the DC magnetization data were recorded while the Fe_5_GeTe_2_ single crystal was warmed up. This resulted in a ZFC magnetization curve. The FC magnetization curve is obtained by lowering the temperature of the Fe_5_GeTe_2_ single crystal from 350 K to 2 K in the presence of an applied magnetic field of 3000 Oe.

As the temperature decreases from 350 K, the magnetization M(T) rises to a markedly high value of magnetization, as shown in Figure 3a, which indicates the transition of the paramagnetic to ferromagnetic phases. It is evident that a plateau transition is occurring as the cooling process continues. However, at low temperatures, there is an obvious bifurcation between the magnetization curves of the ZFC and the FC along the c-direction, exhibiting an extreme value of approximately 108 K in the ZFC and 103 K in the FC processes. In the magnetically ordered state of ferromagnetic materials, this behavior is very extraordinary. A possible interpretation is that either the samples may undergo a structural phase transition [23] or other magnetic phase transitions in this temperature regime. The crystal structure of Fe_5_GeTe_2_ was measured at 103 K, 108 K, and 298 K by variable-temperature X-ray diffraction. The results of this measurement demonstrate that Fe_5_GeTe_2_ does not undergo a structural phase transition around 103 K and 108 K, as shown in Figure 4. Furthermore, this phenomenon may be due to the ferromagnetic domain-wall pinning effect [24], spin-glass freezing behavior [25,26,27,28], the antiferromagnetic correlation between the local spins and itinerant electrons [29], or other kinds of magnetic inhomogeneity.

Spin glasses are magnetic systems in which the interactions between the magnetic moments are ‘in conflict’ with each other due to some frozen-in spin disorder. Therefore, it is impossible to establish any conventional long-range order (neither ferromagnetic nor antiferromagnetic type) [30]. The effect of ‘disorder’ has been employed for the definition and the characteristics of spin glass states. A spin glass system is defined as a system in which the magnetic moments exhibit disordered orientation. In such disordered magnetic systems, the magnetic moments frequently exhibit competing relationships between antiferromagnetic and ferromagnetic interactions. Here, we demonstrated that Fe_5_GeTe_2_ potentially exists in a spin glass state below the freezing temperature based on the observed correlation between the *M*_ZFC_ and *M*_FC_ relationships. Two key characteristics of spin glass systems are magnetic frustration and magnetic disorder. The sample was cooled in fields of zero and 3 k Oe, after which magnetization measurements were taken upon warming at 3 k Oe. *M*_ZFC_ falls below *M*_FC_ below the freezing temperature *T*_f_ (~98 K), which is potentially a typical behavior observed in spin-glass materials. The distribution of the magnetic ion concentration in the magnetic system is inhomogeneous, which in turn gives rise to inhomogeneous interactions between the magnetic moment spins. Furthermore, this will result in antiferromagnetic interactions being present in the system at all times, which will cause ferromagnetic frustration. The formation of a ferromagnetic region is initiated by a phase transition occurring first in the region of stronger magnetic interactions. At elevated temperatures, the level of ferromagnetic frustration is relatively minimal. However, as the temperature decreases, the frustration effect gradually increases. When the level of frustration is sufficiently high, the energy required for the system to form a long-range magnetic order is greater than that needed for the formation of a spin-glass state. As a result, the system will gradually evolve into a spin-glass state [31]. It is also noteworthy that the FC and ZFC curves exhibit two intersections in the low-temperature region, at 98 K and 118 K, which differ from the previously reported spin-glass states [27,28]. The anomalous behavior observed between 98 K and 108 K requires further investigation, specifically through the measurement of the AC magnetic susceptibility and the magnetic relaxation effects.

As shown in Figure 3b, the first derivatives of the magnetization with respect to the temperature of Fe_5_GeTe_2_ under both field-cooled (FC) and zero-field-cooled (ZFC) conditions reveal three distinct extremes. In the case of the FC condition, the temperatures of extremes are 117 K, 276 K, and 302 K, while in the case of the ZFC condition, the corresponding temperatures are 120 K, 310 K, and 334 K. The ferromagnetic transition temperature (*T*_C_) of the current Fe_5_GeTe_2_ crystal was approximated by examining the first derivative of the magnetization with respect to temperature as a function of temperature. The resulting *T*_C_ was determined to be 302 K, as shown in Figure 3b; this is close to the value previously reported [13]. In our system, the *T_C_* of Fe_5_GeTe_2_ single crystals is 302 K, which is higher than the *T*_C_ of the Fe_3_GeTe_2_ single crystal (*T*_C_ = 220 K) [32] and the *T*_C_ of the Fe_4_GeTe_2_ single crystal (*T*_C_ = 270 K) [33]. The elevated *T*_C_ could be attributed to the enhanced Fe-Fe exchange interaction among neighboring atoms as the consequence of the increased incorporation of Fe into the system, thereby significantly augmenting the number of Fe neighbors [18].

Figure 3c shows the temperature dependence of the inverse susceptibility 1/χ. The data in the high-temperature range were well fitted by the modified Curie–Weiss law:(1)$$\chi =\chi_{0}+\frac{C}{T-\theta }$$

In this context, χ_0_ represents the temperature-independent susceptibility, C denotes the Curie–Weiss constant, and θ signifies the Weiss temperature. The fitting process yielded the following parameter values: C = 8.32534 emu K/mol, θ = 305.06639 K, and χ_0_ = 0.02728 emu/mol for H || *c*. The results indicate that at elevated temperatures, χ(T) displays a nearly isotropic paramagnetic behavior. The Curie–Weiss temperature (θ = 305.06639 K) was obtained from the Curie–Weiss fit, which is similar to that of the Curie temperature (*T*_C_ = 302 K), which was estimated based on the temperature dependence of the first derivative of the magnetization with respect to temperature. A positive θ indicates that the ferromagnetic interaction between the Fe atoms is the dominant interaction between the layers.

The temperature dependence of χ^−1^ derived from the *M*(T) curves for the Fe_5_GeTe_2_ crystal under zero-field-cooled (ZFC) conditions is shown in Figure 3d. The downturn in inverse susceptibility (χ^−1^) as a function of temperature above *T*_C_ is considered to be a signature of Griffiths singularity [33]. In the high-temperature regime, the inverse susceptibility (χ^−1^) exhibits a linear relationship with temperature, according to the Curie–Weiss (CW) law. The Griffiths phase (GP) is the transitional phase between the FM and PM phases, when inverse susceptibility [χ^−1^(T)] depicts a decline as a function of temperature. The occurrence of Griffith phases was evaluated for both Tb_5_Si_2_Ge_2_ [34] and La_1−x_Sr_x_MnO_3_ [35] based on the observation that the inverse-susceptibility [χ^−1^(T)] curves exhibit a deviation from linear dependency, characterized by a downward bend. In this work, a pronounced downward bend in χ^−1^(T) above the critical temperature (*T*_C_) (Inset Figure of Figure 3d) is observed, which indicates the appearance of the Griffiths phase (GP). In addition, the magnetic field plays a significant role in influencing the Griffiths phase, and the presence of an externally applied field can effectively suppress the Griffiths phase [35].

In order to further elucidate the anisotropic magnetic properties of the synthesized Fe_5_GeTe_2_ crystal, isothermal magnetization measurements were conducted at temperatures of 20, 100, 150, 200, and 300 K, and the resulting field dependences are presented in Figure 5a. In the magnetically ordered state at 20 K, the magnetization of Fe_5_GeTe_2_ crystals saturates in the *c*-direction with negligible hysteresis losses, indicating a soft ferromagnetic behavior similar to that of Fe_3_GeTe_2_ [36]. The inset in Figure 5a presents the isothermal magnetization (*M*-*H*) as a function of the magnetic field *H* at 20 K. From this data, we determine a saturation magnetization (*M*_sat_) of 4.7 *μ_B_*/Fe at 20 K. This saturation magnetization is higher than those of Fe_5_GeTe_2_ (2.1 *μ_B_*/Fe at 3 K, H//ab) [18], Fe_3_GeTe_2_ (1.625 *μ*_B_/Fe) [9], and Fe_4_GeTe_2_ (1.8 *μ*_B_/Fe) [33] single crystals. Furthermore, as the temperature increases, the saturating magnetization demonstrates a gradual decrease.

Based on the Landau mean field theory [37], the M-H in the vicinity of the Curie temperature can be expressed as follows:(2)$$a\left(T\right)+b\left(T\right)M^{2}=\mu_{0}\frac{H}{M}$$

Consequently, it can be reasonably deduced that the Arrott plot (M^2^-H/M) should ideally form a straight line with an intercept that approaches zero as the temperature T nears the Curie temperature *T*_C_. Additionally, the Curie temperature (*T*_C_) can be accurately determined from the Arrott plot. The reliability of this methodology has been substantiated in Fe_3_GeTe_2_ crystal [9] and Ni_3_Al [38]. Figure 5b shows the M^2^ vs. H/M plots for Fe_5_GeTe_2_ at various temperatures for H || *c*. We find that the positive slope of the M^2^ vs. H/M plots near *T*_C_ suggests that the Fe_5_GeTe_2_ crystal undergoes a second-order magnetic transition at *T*_C_. However, M^2^ does not show a linear relationship with H/M over the whole range of temperatures; instead, it shows a curvature. However, at the high-field region, all the Arrott plots show a linear relation. In this context, the M^4^ vs. H/M plot is very useful to determine *T*_C._ This has been confirmed in Fe_x_Co_1−x_Si [39] and Fe_3_GeTe_2_. The linear correlation observed between M^4^ and H/M can be elucidated by Takahashi theory, which incorporates the influence of the critical thermal amplitude of spin fluctuations under the applied magnetic field. Therefore, we replotted the isothermal magnetization curves in the form of M^4^ vs. H/M, as shown in Figure 5c. From the M^4^ vs. H/M plot, *T*_C_ is determined to be greater than 300 K for Fe_5_GeTe_2_, which is consistent with the *T*_C_ (=302 K) we obtained earlier.

## 5. Summary

In summary, this work presents an experimental study of the structure, DC magnetization, and isothermal magnetization data of Fe_5_GeTe_2_ single crystals grown by chemical vapor phase transport. Fe_5_GeTe_2_ single crystal has a trigonal crystal structure with space group R3m (No. 160). DC and isothermal magnetization studies of the Fe_5_GeTe_2_ crystals show that a ferromagnetic (FM) to paramagnetic (PM) transition occurs at 302 K (*T*_C_). In the low-temperature region, Fe_5_GeTe_2_ crystal has a potential spin glass state with a freezing temperature (*T*_f_) of 98 K. However, there are two intersections in the low-temperature region of the FC and ZFC curves, which differ from the previously reported spin glass states. In the high-temperature region, Fe_5_GeTe_2_ exhibits non-linear magnetization due to the Griffiths phase.

## Figures and Tables

**Figure 1 nanomaterials-15-00019-f001:**
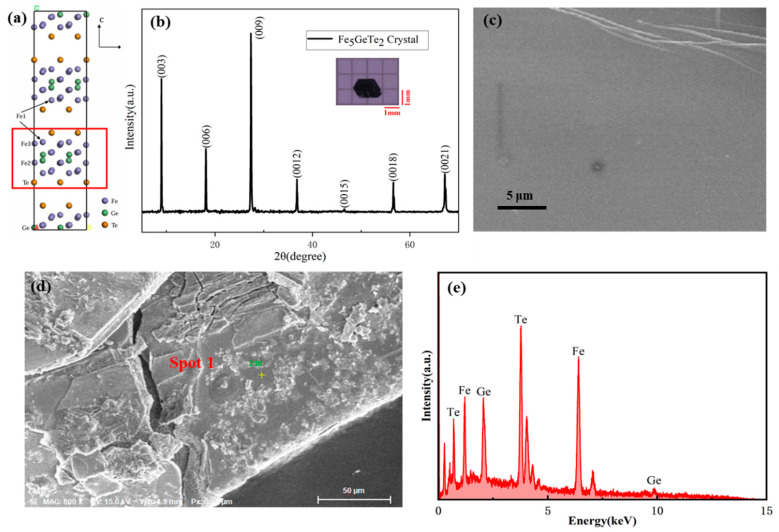
(**a**) Crystal structure of Fe_5_GeTe_2_. Unit cell projected along [001], with Fe_5_GeTe_2_ layers separated by the vdW gap between adjacent Te sublayers. (**b**) X-ray diffraction pattern of Fe_5_GeTe_2_ single crystal while the X-rays were incident on a flat surface of the crystal (the inset shows the photo of the single crystal; the black grid is 1 mm × 1 mm inside). (**c**) The SEM image of the Fe_5_GeTe_2_ single crystal. (**d**,**e**) Typical EDX measurement result of the Fe_5_GeTe_2_ single crystal.

**Figure 2 nanomaterials-15-00019-f002:**
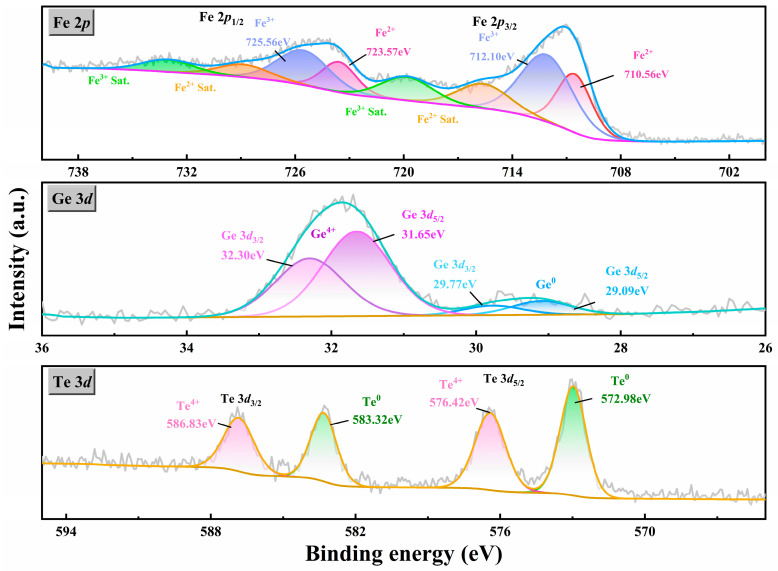
The XPS analysis of the Fe_5_GeTe_2_ single crystal is focused on the Fe *2p*, Ge 3*d*, and Te *3d* elements. A detailed examination was conducted of the composition and valence states of the elements.

**Figure 3 nanomaterials-15-00019-f003:**
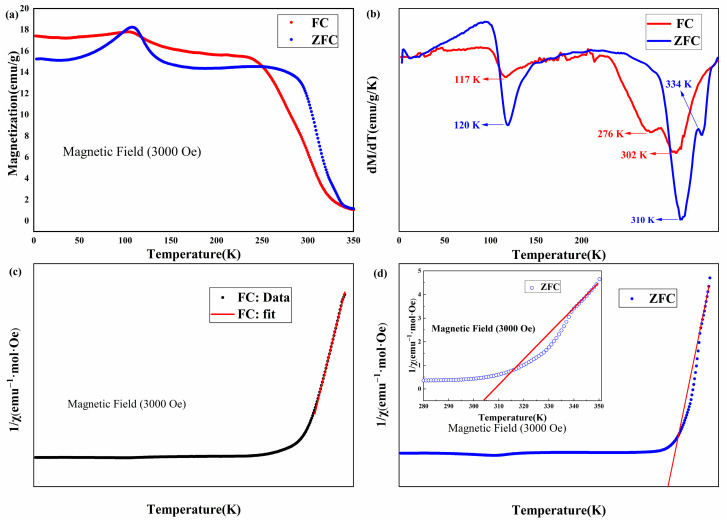
(**a**) M-T curves of Fe_5_GeTe_2_ single crystal at 3000 Oe magnetic field; (**b**) dM/dT-T curves of crystal under field-cooled-cooling (FC) and zero-field-cooled (ZFC); (**c**) The curve of the inverse magnetic susceptibility of the Fe_5_GeTe_2_ crystal under field-cooled (FC) condition with respect to temperature. The red line represents the fit according to the modified Curie–Weiss law; (**d**) The curve of the inverse magnetic susceptibility of the Fe_5_GeTe_2_ crystal under zero-field-cooled (ZFC) conditions with respect to temperature. The red line is the guide to eyes to note the deviation. The inset of Figure 3d illustrates the inverse magnetic susceptibility curve as a function of temperature for a narrower temperature range near *T*_C_.

**Figure 4 nanomaterials-15-00019-f004:**
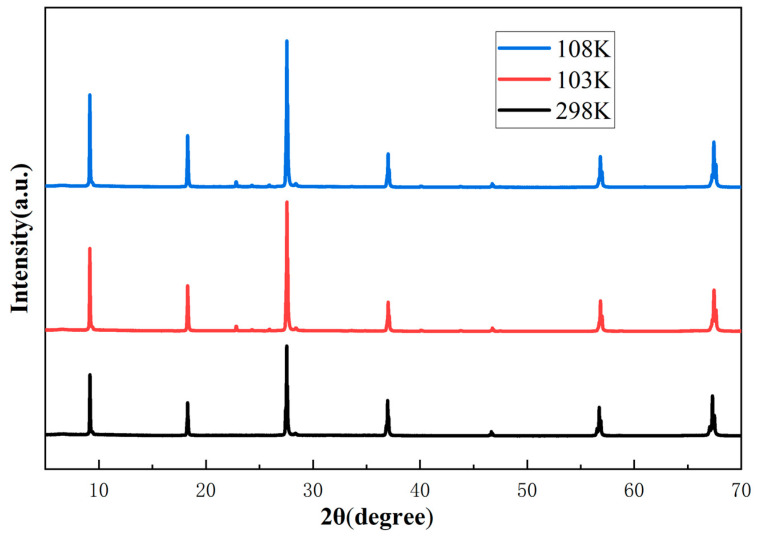
Variable-temperature X-ray diffraction pattern of Fe_5_GeTe_2_ single crystal at 103 K, 108 K, and 298 K.

**Figure 5 nanomaterials-15-00019-f005:**
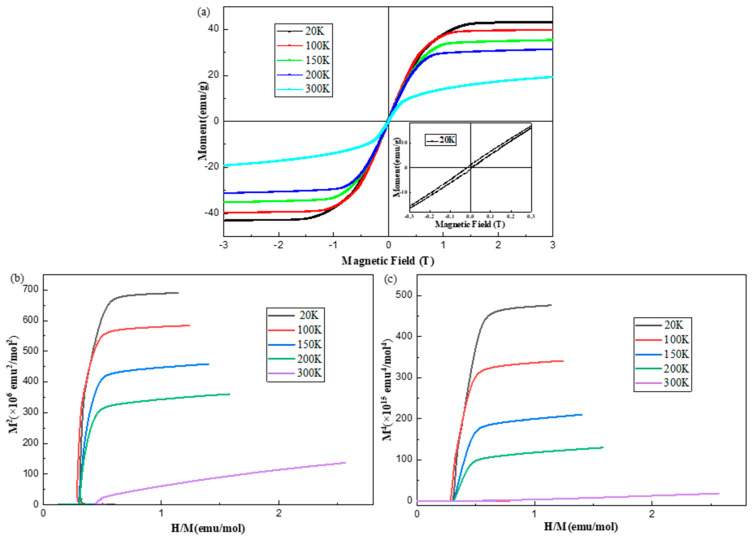
(**a**) The isothermal magnetization curves for the Fe_5_GeTe_2_ crystal were measured at temperatures of 20, 100, 150, 200, and 300 K, with the magnetic field applied along the crystallographic *c*-axis direction. The inset in Figure 5a provides a more detailed illustration of the hysteresis loop of magnetization (M-H) as a function of the magnetic field H at 20 K; (**b**) M^2^ vs. H/M plots (Arrott plot) for Fe_5_GeTe_2_ at various temperatures with H||*c*; (**c**) M^4^ vs. H/M plots for Fe_5_GeTe_2_ at various temperatures with H||*c*.

## Data Availability

The original contributions presented in the study are included in the article and Appendix A; further inquiries can be directed to the corresponding author.

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
