# Peer review of "Spin Glass State and Griffiths Phase in van der Waals Ferromagnetic Material Fe5GeTe2"

_nanomaterials, 2024, doi:10.3390/nano15010019_

Round 1

Reviewer 1 Report

Comments and Suggestions for Authors

The manuscript ''Spin glass state and Griffiths phase in van der Waals ferromag
netic material Fe5GeTe2'' by He et al. presents an experimental report on the growth and characterization of bulk van der Waals magnetic system Fe5GeTe2. The authors describe the magnetic properties of the crystals in terms of spin glass behavior and Griffiths phase. There are a few comments:

1. The authors should give the temperature dependent XRD in the main text instead of the supplementary materials.

2. The authors report presence of elemental Ge in the crystals as determined from the XPS measurements. How can the authors be sure that the unreacted Ge in the crystal does not influence the observed properties. Please give a justification.

3. In van der Waals materials such as the one reported here, self intercalation of the transition metal ion has been reported, for example in CrTe2. Did the authors check if there is any evidence of self intercalation? Can the self intercalation of Fe in the ctystal explain the observed behavior? The authors should provide experimental evidence to support their arguments.

4. The proposal of the Griffith's phase is not clear in the text. A better explanation must be provided to support the conclusions of the authors.

I would recommend a major revision and resubmission for further consideration.

Comments on the Quality of English Language

The English is satisfactory but room for improvement is there.

Reviewer 2 Report

Comments and Suggestions for Authors

The authors synthesized high-quality samples of the material Fe5GeTe2, which is currently being intensively studied. The interest in it is due to the fact that it is a two-dimensional ferromagnet with a high Curie temperature. The authors conducted detailed studies of the structural and magnetic properties of the samples. Interesting results were obtained from the M(T) and M(H,T) data, which the authors interpret within the framework of modern concepts of magnetism.

I believe that these results are new, will arouse interest in the scientific community and therefore the work can be published in the journal Nanomaterials.

I have only minor comments:

1. It would be better to use the original work of A. Geim et al. as reference 1.

2. Figure 4a. What is the point of showing the same M(H) curve in the inset on the same scale as in the main panel?

Reviewer 3 Report

Comments and Suggestions for Authors

 .  The magnetic properties vs. H and T plots look hand-drawn to be rendered authentic.

Please include a couple of plots with the actual data (subtract off the substrates) so that the results can be evaluated.

.  The manuscript is nearly entirely based on observation, but scientific underpinning is missing, just one example, the manuscript mentions "disorder", but the effect must be discussed.

. SG transition lacks the frequency dependence to support it.

. To a lesser extent, explain why SG and ferromagnet-paramagnetic transitions are important in this compound.

Round 2

Reviewer 1 Report

Comments and Suggestions for Authors

The authors have replied to all the comments and questions posed. I am satisfied with the revisions and the manuscript can be accepted for publishing. 

Reviewer 3 Report

Comments and Suggestions for Authors

the authors answered my questions, I approve their answers